# Single-cell transcriptomics of a dynamic cell behavior in murine airways

**Sheldon JJ Kwok[1,2†‡], Daniel T Montoro[3,4†§], Adam L Haber[5†], Seok-Hyun Yun[1,2*], Vladimir Vinarsky[6,7*#]**

[1]Harvard Medical School and Wellman Center for Photomedicine, Massachusetts General Hospital, Cambridge, United States; [2]Harvard-MIT Health Sciences and Technology, Massachusetts Institute of Technology, Cambridge, United States; [3]Broad Institute of MIT and Harvard, Cambridge, United States; [4]Department of Systems Biology, Harvard Medical Schoo, Boston, United States; [5]Department of Environmental Health, Harvard T. H. Chan School of Public Health, Boston, United States; [6]Center for Regenerative Medicine, Massachusetts General Hospital, Boston, United States; [7]Division of Pulmonary and Critical Care Medicine, Department of Medicine, Massachusetts General Hospital, Boston, United States

**\*For correspondence:**
syun@hms.harvard.edu (S-HY);
vvinarsky@gmail.com (VV)

[†]These authors contributed equally to this work

**Present address:** [‡]LASE Innovation Inc, Cambridge, United States; [§]TenSixty Biosciences, Inc, Cambridge, United States; [#]Vertex Pharmaceuticals Incorporated, Boston, United States

## Abstract

Despite advances in high-dimensional cellular analysis, the molecular profiling of dynamic behaviors of cells in their native environment remains a major challenge. We present a method that allows us to couple the physiological behaviors of cells in an intact murine tissue to deep molecular profiling of individual cells. This method enabled us to establish a novel molecular signature for a striking migratory cellular behavior following injury in murine airways.

## Editor's evaluation

This study presents a useful combination of live cell imaging with single-cell transcriptomic analyses. This is a first step to further expanding the description of cellular heterogeneity, including cellular behavior as well as gene expression profiles. The method, with more technical improvements, will be of interest to researchers who study dynamic changes in cell morphology and gene expression.

## Introduction

Cells in a living organism are dynamic entities, changing their characteristics over space and time and constantly interacting with the host and pathogens. The ability to obtain such information and link it to detailed molecular phenotypes of the cells would be highly useful for biomedical investigations but has been underappreciated. Here, we present a method that allows us to characterize complex physiologic behaviors of cells in an intact tissue and then perform live imaging-guided sequencing of the same cells. We validate this approach using a regeneration model of airway tissues and demonstrate how this method leads to new biological findings.

There is a pressing need for a comprehensive understanding of cellular behaviors in the lung, the site where aberrant cellular behavior has been linked to asthma (*Kim et al., 2012*; *Park et al., 2015*) pulmonary fibrosis (*Fukumoto et al., 2016*), and viral infections including influenza and coronaviruses (*Kumar et al., 2011*). Single-cell RNA-sequencing (scRNA-seq) has emerged as a precise way to define cell type and cell state, and new techniques are being developed to determine the spatial distribution of sequenced cells in tissues (*Marx, 2021*). However, the molecular pathways that drive the cellular behavior in situ continue to be inferred from time-lapse tissue sampling or transcriptional

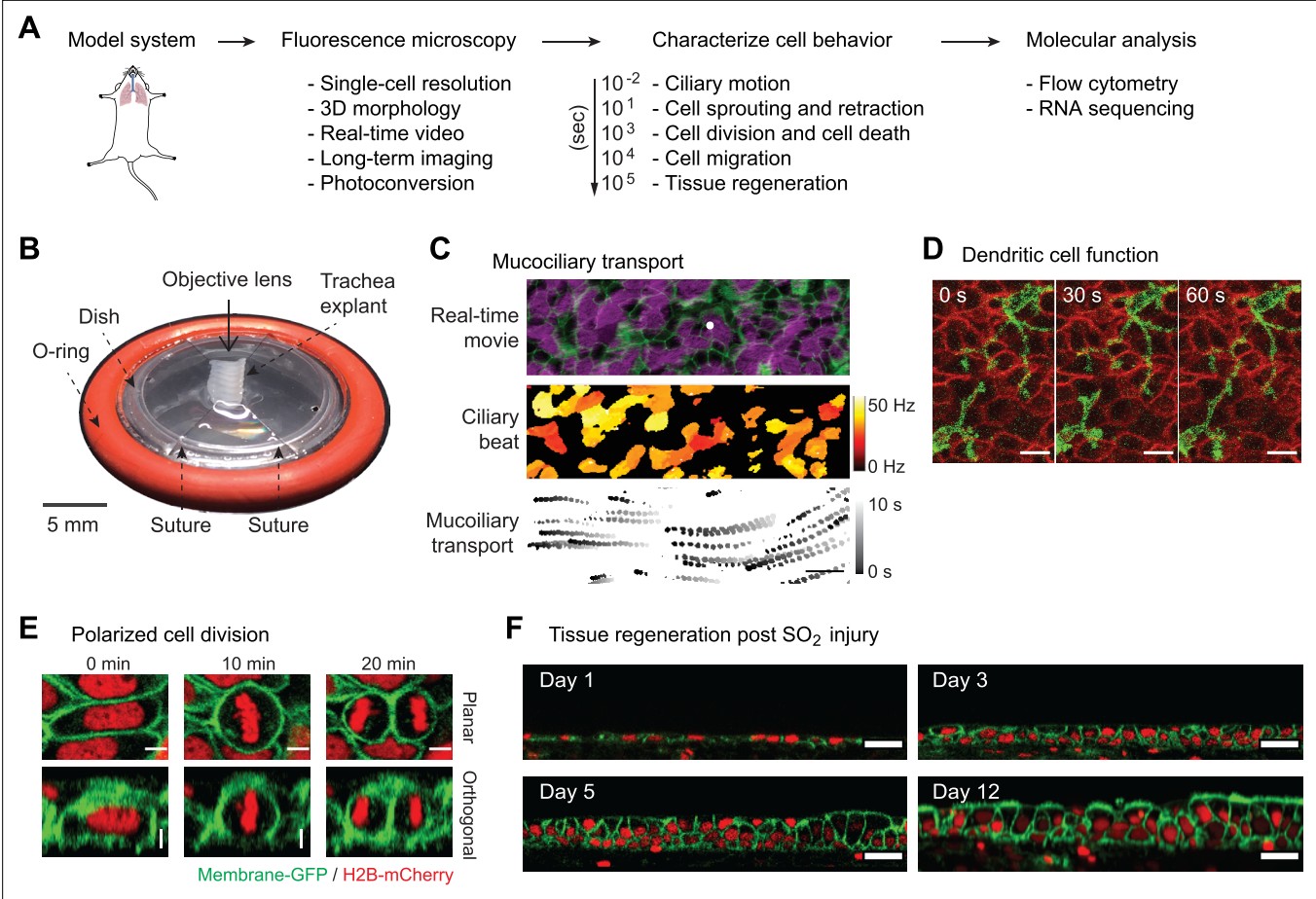

**Figure 1.** Platform for live imaging of airway tissue. (**A**) Behavioral transcriptomics workflow, starting with imaging, followed by image analysis to characterize cellular behavior over different time frames, leading to single-cell applications. (**B**) Airway tissue (trachea) is explanted from a mouse and affixed to a custom platform for long-term air-liquid-interface (ALI) culture and imaging. The platform enables both time-lapse microscopy and downstream single-cell applications. (**C**) Imaging and image analysis of ciliary beating and mucociliary transport 1 μm spherical beads. (**D**) Intraepithelial dendritic cells (CD11cCre-MTMG) grow and retract dendrites in real time; scale bar = 5 μm (**E**) Selected snapshots of cell division during regeneration post-sulfur dioxide ($SO_2$) injury. Epithelial cell divides along its long axis during regeneration (Hertwig's rule); scale bar = 5 μm. (**F**) Long-term ALI culture enables imaging of tissue regeneration post-$SO_2$ injury over >12 days. ALI culture enables regeneration of entire epithelial thickness; scale bar = 20 μm. Green = membrane GFP; red = nuclear-tdTomato.

The online version of this article includes the following figure supplement(s) for figure 1:

**Figure supplement 1.** Platform for live imaging of airway tissue.

kinetics (*La Manno et al., 2018*). Moving beyond inference requires coupling visualized in situ cell behavior with deep molecular profiling of visualized cells.

Live cell imaging is an established technique for capturing morphology and cellular dynamics such as cellular migration during skin regeneration (*Park et al., 2019*; *Park et al., 2017*), but imaging in the lung remains challenging due to difficult access and the constant motion of the respiratory system. Additionally, molecular information that accompanies live imaging is largely limited to a few fluorescent reporters. Prior attempts to link deep molecular profiling with live imaging have relied on imaging dissociated cells (*Lane et al., 2017*; *Yuan et al., 2018*), cell monolayers (*Hu et al., 2020*), organoids (*Konen et al., 2017*) rather than on cell behaviors in their native tissue environment.

Our aim is to bridge the divide between two powerful methodologies - cell behavioral observation through live imaging and transcriptional profiling through single-cell sequencing, ultimately allowing the identification of a transcriptional signature that corresponds to that cell behavior. We present an important step forward toward linking dynamic cell behaviors with single-cell transcriptomics.

# Results

We describe a novel approach to linking live tissue imaging with single-cell profiling (*Figure 1a*). In order to visualize the airway epithelium at high resolution over days, we explant a mouse trachea and secure it in a custom imaging platform without tissue submersion. This approach minimizes sample movement during imaging, maintains a constant supply of nutrients from below the explant, and preserves an air-liquid interface (ALI), which is required for the maintenance of the normal cellular architecture of the airway epithelium (*You et al., 2002*; *Figure 1a* and *Figure 1—figure supplement 1a*). This platform allows imaging of common and rare cell types in the airway epithelium at high resolution in their native environment (*Figure 1—figure supplement 1b*). The explant culture also allows an uninjured tracheal epithelium to survive with its native cellular anatomy for weeks with daily high-resolution imaging (*Figure 1—figure supplement 1c*).

Discernable cell behaviors have a broad time scale, ranging from milliseconds to days. Thus, we imaged a wide range of cellular behaviors, from rapid fluctuations of ciliary beating and directional mucociliary transport over milliseconds to wholescale regeneration of the airway epithelium, which occurs over days after the injury (*Figure 1, c to f*, ). Furthermore, this method allows single-cell level registration within tissues that are live-imaged and subsequently fixed and stained, which enables a unique comparison between live fluorescence cellular patterns and immunostains that describe cell identity and function (*Figure 1—figure supplement 1d*).

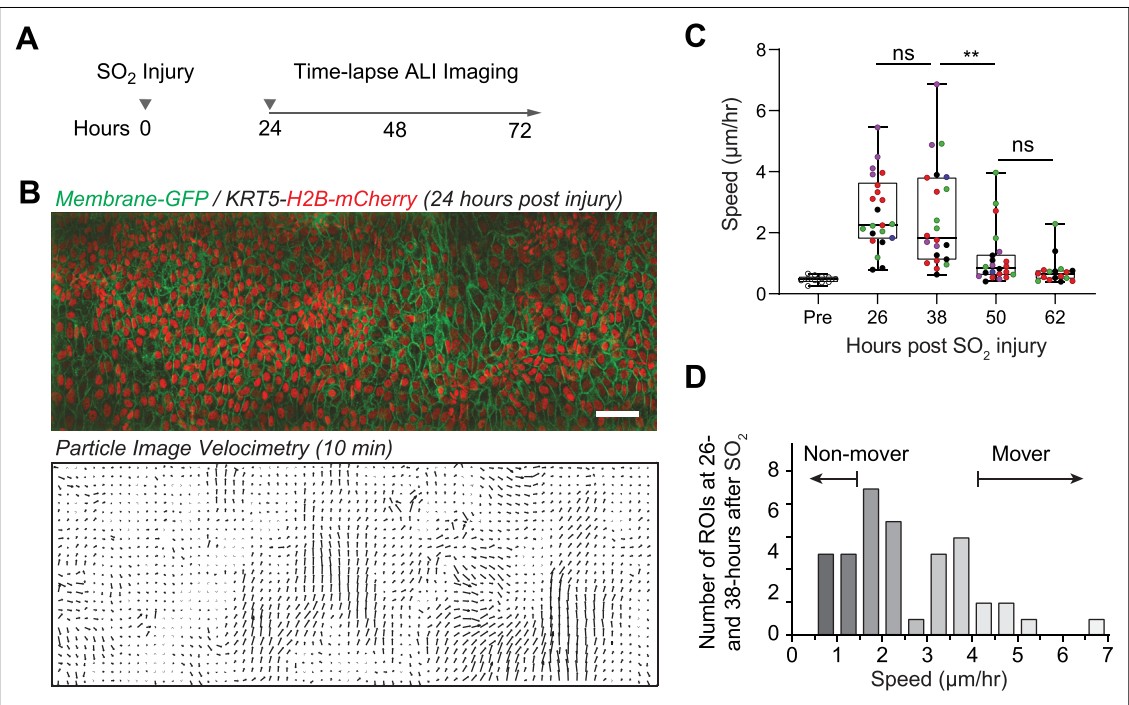

**Figure 2.** Live imaging enables observation of the movement of regenerating airway epithelial cells. (**A**) Experimental design: tracheas are explanted 24 hr post-sulfur dioxide ($SO_2$) injury for continuous time-lapse imaging. (**B**) Two-photon imaging of trachea epithelium from membrane-GFP, KRT5-H2B-mCherry transgenic mouse. Top image is a stitch of three areas. Bottom image shows displacement vectors over 10 min computed using particle imaging velocimetry (PIV). Scale bar = 50 μm. (**C**) Computed speed of epithelial cells measured at different time points post-$SO_2$ injury in 22 independent regions from a total of five mice at four different time points (mouse origin is color-coded). Box and whisker plots are superimposed. A two-way ANOVA was run to examine the effect of time post-$SO_2$ injury and different mice on the mean speed determined by PIV. There were 22 ROIs analyzed from five mice over four time points. There was a significant interaction between time and the mean speed, $F(2.219, 42.91) = 16.12$, $p<0.0001$, but no significant difference between mouse and mean speed, $F(4,17) = 2.193$, $p=0.113$. A Tukey post-hoc test revealed significant pairwise differences between 26 and 50 hr, 26 and 62 hr, 38 and 50 hr, as well as 38 and 62 hr. **$p<0.01$. (**D**) Frequency distribution of injury-induced cell movements measured at 26- and 38 hr after injury identifies 'mover' and 'non-mover' regions.

The online version of this article includes the following figure supplement(s) for figure 2:

**Figure supplement 1.** Epithelial regeneration after sulfur dioxide injury ex vivo.

**Figure supplement 2.** Live imaging enables quantitative analysis of epithelial cell movement over time.

**Figure supplement 3.** Live imaging with high temporal resolution.

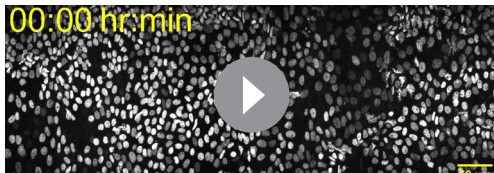

**Video 1.** Timelapse two-photon imaging at 20 min intervals of a regenerating airway epithelium starting at 24 hr after sulfur dioxide injury of a KRT5-H2B-mCherry transgenic mouse. The video was composed by stitching maximum intensity projections of the H2B-mCherry signal from three adjacent areas.

https://elifesciences.org/articles/76645/figures#video1

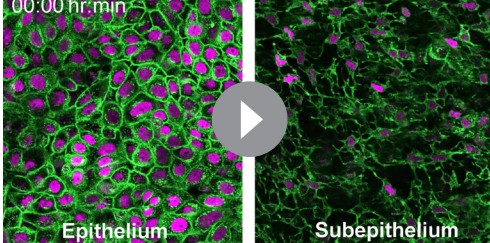

**Video 3.** Timelapse two-photon imaging at 10 min intervals of a regenerating airway epithelium starting at 24 hr after sulfur dioxide injury of a Membrane-GFP/CAGs-H2B-mCherry transgenic mouse. The video was composed of single optical sections through the epithelium at 5 µm above the basement membrane (left) and through the mesenchyme at 5 µm below the basement membrane (right).

https://elifesciences.org/articles/76645/figures#video3

Remarkably, this airway imaging platform faithfully recapitulates and captures cellular dynamics of epithelial regeneration from native basal stem cells after an extensive epithelial injury induced by sulfur dioxide ($SO_2$) (*Figure 1f* and *Figure 2—figure supplement 1*). In the first 5 days, the basal cells divide, increasing cellular density and reforming the pseudostratified epithelium. In the next 5–10 days, the epithelium differentiates, leading to the restoration of the full epithelium, including the regeneration of ciliated cells. Complete regeneration requires an air-liquid interface (*Figure 1f*). Overall, this murine trachea explant ALI culture retains the nearly complete 3D organization and microenvironment of the basal progenitor cells and, therefore, offers a unique model to study organ physiology and regeneration outside of the body.

Continuous time-lapse imaging of the airway epithelium for up to 80 hr after injury (*Figure 2a* and *Figure 2—figure supplement 2a* and *Video 1*) demonstrated changes in cellular architecture over time, including an increase in the average cellular density and epithelial thickness, without apparent phototoxicity (*Figure 2—figure supplement 2a*). Imaging of trachea explant controls from uninjured mice over 19 hr revealed no cellular displacement in the airway epithelium (*Video 2*). We examined cell movement using single-cell tracking following segmentation of cell nuclei and particle image velocimetry (PIV) of non-segmented images (*Figure 2—figure supplement 2b,c*). These analyses revealed a variety of regeneration cellular dynamics inaccessible without live imaging. For example, Hertwig's Rule predicts that a cell division plane is perpendicular to the long axis of the cell during the preceding interphase (*Minc and Piel, 2012*). This was established in plants (*Besson and Dumais, 2011*) and developing simple model organisms (*Aigouy et al., 2010*; *Concha and Adams, 1998*; *Tsou et al., 2003*) but has never been probed in an adult regenerating tissue. We found that the long axis in most cells predicts the cell division axis, while the axis of cellular movement prior to cell division does not (*Figure 1e* and *Figure 2—figure supplement 3a,b*).

We also found a surprising degree of heterogeneity of collective cellular migration during regeneration throughout the injured airways. In regions that demonstrated rapid cellular movement after an injury, the movement peaked at 26–38 hr after $SO_2$ injury and the speed declined significantly in most regions by 50 hr after injury (*Figure 2c*). There was a significant interaction between time and the mean speed, but no significant difference between the mouse and mean speed (*Figure 2c*). Variable migratory behavior of airway epithelial cells has been observed in cell culture models (*Kim et al., 2020*; *Park et al., 2015*) but not previously in an intact regenerating airway tissue. We found that the frequency

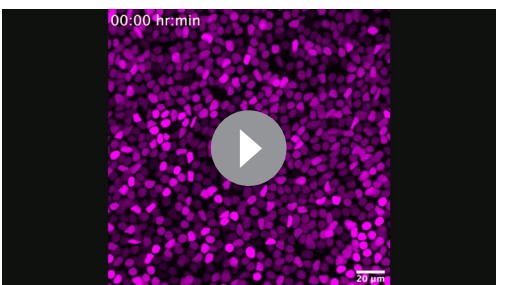

**Video 2.** Timelapse two-photon imaging at 40 min intervals of an uninjured airway epithelium of a ROSA-nuclear-tdTomato (nT-nG) transgenic mouse. The video is composed of maximum-intensity projections of the nuclear tdTomato signal from optical sectioning through the airway epithelium.

https://elifesciences.org/articles/76645/figures#video2

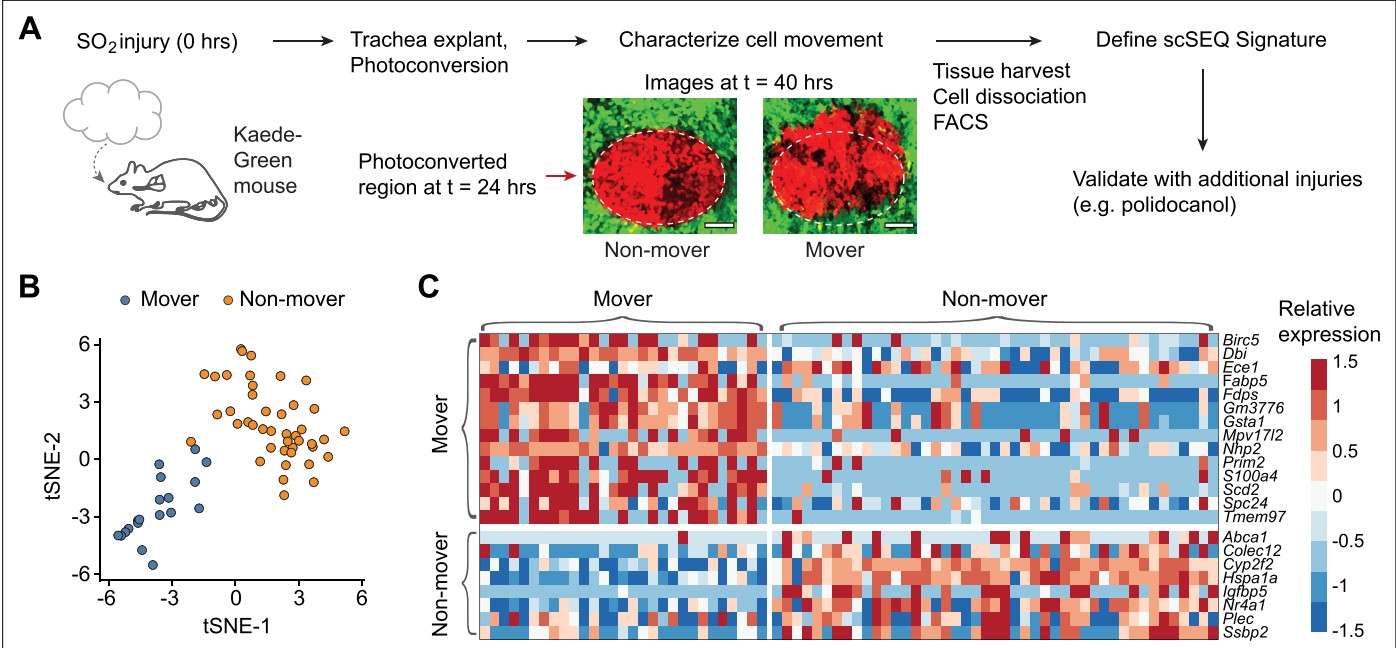

**Figure 3.** Transcriptionally distinct moving (M) and non-moving (NM) cells coordinate early airway epithelial regeneration across multiple injury types. (**a**) Experimental design: tracheas are explanted 24 hr post-sulfur dioxide (SO₂) injury (24 hpi) for continuous time-lapse two-photon imaging. Distinct cellular phenotypes are observed and labeled by photo-conversion for subsequent isolation and transcriptional analysis by full-length single-cell RNA-sequencing. Scale bar = 100 µm (**b**) Unsupervised clustering of regenerating cells partitions mover and non-mover cell phenotypes. (**c**) Heatmap of transcriptional signatures of mover and non-mover cells.

The online version of this article includes the following figure supplement(s) for figure 3:

**Figure supplement 1.** Unsupervised clustering of single-cell RNA-sequencing (scRNA-seq) data identifies club and basal cells.

**Figure supplement 2.** Transcriptional programs distinguish moving (M) and non-moving (NM) basal cells.

distribution of cellular speed in different regions at 26–38 hr demonstrated large variability ranging from 'non-mover' regions (<1.5 µm/hr) to 'mover' regions (>4 µm/hr) (*Figure 2d*). Furthermore, videos with higher temporal resolution revealed that the cells with much slower movements in the 'non-mover' regions had no directional preference and oscillated in place, whereas the 'mover' regions with faster cellular movements were more uni-directional with packs of cells migrating in waves (*Figure 2— figure supplement 3c,d* and *Video 3*).

Distinct subsets of cells have been theorized to contribute to the regeneration process (*Pardo-Saganta et al., 2015*; *Tadokoro et al., 2014*), but it is unclear whether these heterogeneous transcriptional cell states reflect gene expression stochasticity or correlate with unique cell behaviors. To determine the molecular signatures of cells with directional movement compared to regenerating non-moving cells, we marked epithelial regions by photoconversion using the Kaede transgenic mouse model, in which cells change from green to red after exposure to violet light (*Tomura et al., 2008*). We photoconverted cells at 24 hr after injury, imaged every 6 hr, screened for 'movers' with >50 µm displacement (>3 µm/hr) at 18 hr after photoconversion, and then isolated photoconverted epithelial cells by FACS for plate-based scRNA-seq (*Figure 3a* and Materials and methods). Dimensionality reduction revealed that cells from a 'moving' region (M) and a 'non-moving' (NM) region cluster separately (*Figure 3b*). Using unsupervised clustering and cell identity signatures (Methods, *Figure 3— figure supplement 1*), we found that nearly all the cells in the M region are basal cells, whereas the NM region contains basal and club cells (*Figure 3c*). We identified the differences in gene expression between the M basal cells and NM basal cells and identified gene signatures that are enriched (FDR <0.05, likelihood-ratio test) either in the M or the NM basal cells (*Figure 3c* and *Figure 3—figure supplement 2*).

We hypothesized that the identified phenotypes are a common feature of injury-induced epithelial regeneration. We examined published data of an independent injury model (*Plasschaert et al., 2018*) and analyzed the prevalence of these signatures during repair after polidocanol injury. As predicted,

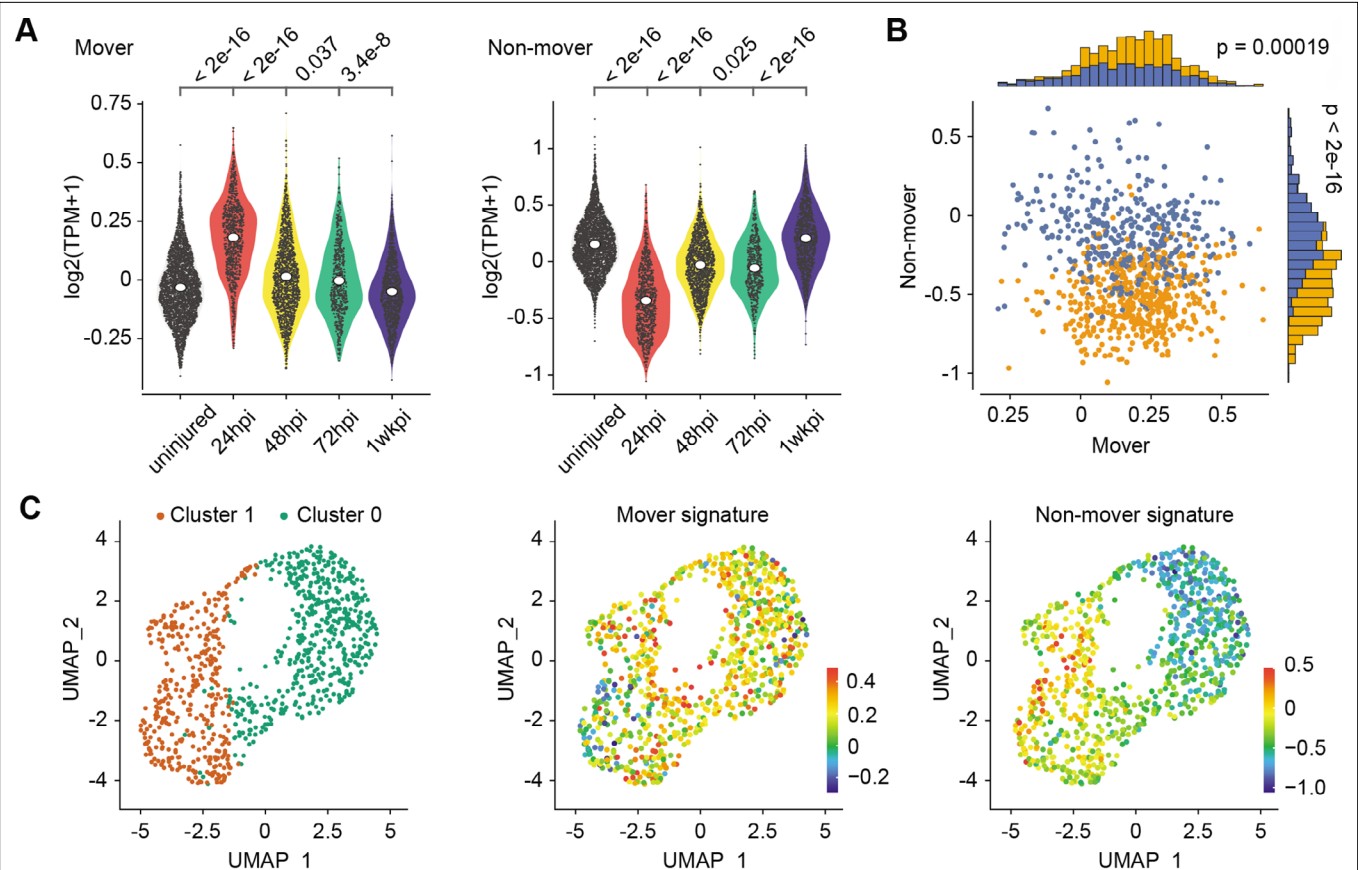

**Figure 4.** Transcriptionally distinct moving (M) and non-moving (NM) cells coordinate early airway epithelial regeneration across multiple injury types. (**A**) Mover and non-mover transcriptional signatures are also enriched in early airway epithelial regeneration 24hours post-injury of an independent murine airway injury induced by polidocanol administration. (**B**) Scoring for mover and non-mover transcriptional signatures in 24hpi regenerating cells following polidocanol treatment partitions cells into two populations. (**C**) Unsupervised clustering of 24hpi regenerating cells yields two cell populations enriched in the expression of mover (cluster 0) or non-mover (cluster 1) signatures. P values from Mann-Whitney U test.

the M basal cell signature is strongly enriched 24 hr post injury (hpi), declines at 48 hpi and 72 hpi, and returns to baseline at 1 week after injury (all $p<10^{-16}$, Mann-Whitney U test, *Figure 4a*). Similarly, the NM signature is decreased at 24 hpi when cell migration is presumed to be active, increases at 48 hpi and 72 hpi when cell migration is presumed to be diminished, and returns to baseline at 1 week post-injury when regeneration is complete (*Figure 4a*). Furthermore, at 24 hpi we found that scoring basal cells using M and NM signatures segregated basal cells into two statistically distinct cell populations (*Figure 4b*), indicating that polidocanol regeneration is likely also characterized by these cell phenotypes. To test this possibility, we used unsupervised clustering (Methods) to define two groups of basal cells at 24 hpi and found that these two populations were indeed separately enriched for the M and NM basal cell signatures (*Figure 4c*), confirming the presence of distinct M and NM basal cells during polidocanol regeneration. Taken together, these findings suggest that distinct M and NM cell behaviors are conserved features of early epithelial regeneration and demonstrate that our live imaging-guided single-cell profiling approach can discover generalizable principles of tissue biology.

## Discussion

The rapid progress in single-cell analysis is enabling the discovery and characterization of transcriptionally heterogeneous cells in diverse tissue contexts (*Lee et al., 2021*; *Ståhl et al., 2016*). However, these methods do not capture the dynamics of cell behaviors that often define the unique biological processes that occur in the tissues. To address this gap, we developed an approach to examine the association of molecular and behavioral phenotypes of cells in their native tissues. We established

an airway organ explant culture that maintains tissue dynamics for an extended length of time and combined this platform with live imaging in order to observe distinct airway cellular behaviors at a broad time scale, spanning cell migration, cell division, and ciliary beating.

To link cell behavior to molecular analysis, we used photoconversion to mark cells for subsequent single-cell genomics analysis. We found that a subpopulation of basal stem cells migrates within the lung during early regeneration. We used recently developed single-cell RNA-sequencing approaches to establish molecular signatures for moving and nonmoving basal cells. These distinct cell signatures were identified across independent lung regeneration models, suggesting that M and NM cell behaviors are conserved cellular features of early epithelial regeneration.

There are important limitations to this approach. For example, the optical requirements of the Kaede photoconversion model limit its application for single cell photoconversion. However, a similar approach can take advantage of different methods of cell labeling and live imaging, including genetic labeling of single cells with fluorescent reporters (*Figure 1—figure supplement 1b*). The requirement for fluorescence is another constraint of this approach, although label-free imaging approaches may be used to visualize the behavior of individual cells in native tissues (*Shah et al., 2022*). Another consideration is that long-term live imaging itself can influence gene expression and cell behavior. Although phototoxicity that leads to photobleaching or dramatic changes in cell integrity can be avoided, subtle changes to gene expression or cell behavior due to live imaging may be difficult to detect (*Magidson and Khodjakov, 2013*).

This report focuses on cellular behavior in response to injury in the airway epithelium. Other epithelial organs such as the cornea (*Park et al., 2019*) and esophagus *Doupé et al., 2012*; *Yokobori et al., 2016* have been cultured on similar platforms, and we anticipate this live imaging-guided single-cell profiling approach can be extended to other tissues to discover underlying principles of heterogeneous cellular behaviors at homeostasis and in disease. Although the explant platform lacks some elements of the in vivo microenvironment of these organs, live imaging of the explant followed by molecular analysis overcomes the costly constraints of spatial and temporal resolution during in vivo live imaging (*Haase et al., 2022*), the type of resolution required to link dynamic cell behaviors with single cell transcriptomics.

## Materials and methods

### Mice

mT-mG (stock no. 007676), nT-nG (stock no. 023035), CAGs-LSL-rtTA3 (stock no. 029617), and Col1a1-tetO-H2B-mCherry (stock no. 014602), CD11cCre (stock no. 007567), and Ascl1nGFP (stock no 012881) mice were purchased from the Jackson Laboratory. Foxj1Cre (*Zhang et al., 2007*), KRT5rtTA (*Diamond et al., 2000*), B1EGFP (*Miller et al., 2005*), Foxj1CreER (*Rawlins and Hogan, 2008*), CC10CreER (*Rawlins et al., 2009*), and Kaede (*Tomura et al., 2008*) lines were previously described. A line of Membrane-GFP (mG) mice was generated by selecting GFP-positive pups of a Foxj1Cre-mTmG male parent (with mT to mG recombination in the sperm) and backcrossing to WT background to eliminate the Cre allele. The mG line without Cre was crossed to nT-nG to generate the 'nT-mG' strain. Mice were maintained in an Association for Assessment and Accreditation of Laboratory Animal Care-accredited animal facility at the Massachusetts General Hospital, and procedures were performed with Institutional Animal Care and Use Committee (IACUC)-approved protocols. Mice of all strains were housed in an environment with controlled temperature and humidity, on 12 hr light-dark cycles, and fed with regular rodent's chow.

### SO$_2$ injury

SO$_2$ injury model was performed as previously described (*Kim et al., 2012*; *Pardo-Saganta et al., 2015*). In brief, mice were exposed to 500 p.p.m. of SO$_2$ for 3 hr 40 min and the trachea was collected 16–24 hr after injury for imaging and explant culture.

### Tracheal explant and tissue ALI culture

Tracheas were dissected, cleared of connective tissue and adjacent organs, and opened longitudinally along the anterior tracheal wall. The tracheas were placed on ice in DMEM/F-12 Media with Primocin (InVivoGen) and 15 mM HEPES until culture. Trachea explants were then sutured onto a silicone o-ring

and placed in a custom-made tissue culture dish over an inverted ALI insert secured in a 60 mm tissue culture dish by PDMS. This approach ensured stability during high-resolution imaging. The media contacted the explant from below through the ALI membrane. The dish was placed in a physiological live imaging chamber ($CO_2$ and temperature-controlled, TokaiHit) on the stage of the two-photon microscope.

## Physiological two-photon imaging

Trachea explant imaging was performed on an Olympus FVMPE-RS multiphoton laser scanning microscope equipped with a MaiTai HPDS-O IR pulsed laser (900 nm for GFP and SHG) and INSIGHT X3-OL IR pulsed laser (1100 nm for tdTomato), using a 25 X water immersion lens (NA 1.05). Explants were imaged at time points as indicated in the Figures. For orthogonal view reconstruction, we scanned the trachea with 0.75 μm Z steps. To reimage the same trachea at high resolution at different time points, landmarks such as cartilage rings and vascularity patterns were used as fiducial marks. These fiducial marks were also used for 2D and 3D registration of different time points.

## Image analysis

4D images (x, y, z, t) were imported into MATLAB and/or ImageJ for image processing and analysis. Because the curvature of the tissue changes over time, we first normalized each 3D image to generate a flat basement membrane. As the SHG signal is maximal at the basement membrane, we computed the z height of the basement membrane across the image after applying a Gaussian blur (typical σ values: 10–25 μm in xy, 1–4 μm in z). This height was subtracted from the original 3D data to level the basement membrane. MATLAB code for flattening the 3D images is publicly available on Github at https://github.com/sheldonjjlase/matlab/, (copy archived at swh:1:rev:230ca1a3c03039ae553d-88c3d3f1286404a34a85) (*Kwok, 2022*). Other image processing steps including brightness and contrast adjustments, background subtraction, photobleaching correction, pseudocoloring, 3D time-lapse registration, and stitching were performed using built-in functions in ImageJ.

Cilia beating was recorded by acquiring time-lapse two-photon images of the epithelial surface at 150 Hz over 200 frames using a resonant galvanometer scanner. To estimate the cilia beat frequency (CBF), we estimated the power spectral density of the fluorescence intensity fluctuations across the image using Welch's method in MATLAB. The peak fluctuation frequency was computed for each pixel across the image corresponding to bright cilia. Mucociliary transport was measured by applying 1 μm fluorescent spherical beads to the epithelial surface and recording their displacement over time after equilibration.

To track individual cells over time-lapse imaging, images were imported into ilastik for segmentation and cell tracking (*Berg et al., 2019*). Briefly, pixels corresponding to nuclei were first classified using manual training and machine learning. Next, individual cells were similarly identified through manual training and machine learning algorithms to classify objects. Finally, classified cells were tracked over time using a conservation tracking algorithm. Segmented and tracked cells were then imported into MATLAB for quantitative analysis, including the computation of individual cell speeds over time.

For the automated estimation of cell speed from time-lapse imaging, we performed automated particle imaging velocimetry (PIV). Image sequences were imported into MATLAB and analyzed using the PIVlab plugin (*Thielicke and Stamhuis, 2014*). A direct Fourier transforms correlation with multiple passes of sizes consisting of 24 μm, 16 μm, and 10 μm was used. This generated displacement vectors arranged in a grid with 10 μm spacing for each sequential pair of images. The average cell speed for each time-point was estimated by computing the average absolute displacement estimated by PIV in each $(10 \times 10)$ μm² region divided by the time between images. To quantify the directionality of cell movement, we computed the circular variance of the displacement vectors generated by PIV analysis.

## Kaede photoconversion

Trachea from Kaede mice (*Tomura et al., 2008*) were explanted 20 hr after $SO_2$ inhalation injury, sutured onto a silicone O-ring, and secured on an inverted ALI insert in media on ice, and placed on the imaging platform of a FV3000 Olympus Laser Scanning confocal microscope. Selected regions were outlined and photoconverted using the 405 nm laser for 2 min, while both disappearances of KaedeGreen and the appearance of KaedeRed were simultaneously visualized using the 488 nm and the 561 nm lasers, respectively.

To identify regions of movement and no movement, we explanted Kaede tracheas 20 hr after SO$_2$ injury, photoconverted distinct regions with a specific shape, and proceeded with timelapse live imaging, screening for regions with significant shape displacement over time (from epithelial movement) (*Figure 3a*). After defining whether a region moved or remained non-moving, we excised a trachea fragment, dissociated the fragment into single cells, and used flow-activated cell sorting (FACS) to isolate photoconverted (KaedeRed) epithelial cells. We then proceeded to single-cell RNA sequencing of cells isolated from moving and non-moving regions.

### Cell dissociation and FACS

Airway epithelial cells were dissociated using papain solution. Tracheal fragments with photoconverted regions were trimmed and incubated in papain dissociation solution and incubated at 37 °C for 2 hr. After incubation, dissociated tissues were passed through a cell strainer and centrifuged, and pelleted at 500 g for 5 min. Cell pellets were dispersed and incubated with Ovo-mucoid protease inhibitor (Worthington Biochemical, cat. no. LK003182) to inactivate residual papain activity by incubating on a rocker at 4 °C for 20 min. Cells were then pelleted and stained with EpCAM–BV421 (1:50; BD Bioscience, #563214) for 30 min in 2.5% FBS in PBS on ice. After washing, cells were sorted by fluorescence (antibody staining, Kaede-Green, and Kaede-Red) on a BD FACS Aria (BD Biosciences) using FACS Diva software, and analysis was performed using FlowJo (version 10) software.

Single cells were sorted into each well of a 96-well PCR plate containing 5 µl buffer. After sorting, the plate was sealed with a Microseal F, centrifuged at 800 g for 1 min and immediately frozen on dry ice. Plates were stored at −80°C and submitted to a core facility for cDNA library generation, amplification, and sequencing.

### Single-cell sequencing and sequence analysis

cDNA was generated from single cells in the 96-well plate using the SmartSeq v4 kits (Takara Bio) using 1/4$^{th}$ volume reactions dispensed using a Mantis dispenser (Formulatrix). Samples were amplified using 18 cycles of PCR. Resulting cDNA was then made into Illumina-compatible libraries using the Nextera XT kit (Illumina Inc). Libraries were sequenced on a NextSeq using a Mid Output 150 cycle kit (Illumina Inc) using 75 bp paired-end reads.

### Pre-processing of plate-based scRNA-seq data

BCL files were converted to merged, de-multiplexed FASTQ files using the Illumina Bcl2Fastq software package v.2.17.1.14. Paired-end reads were mapped to the UCSC mm10 mouse transcriptome using Bowtie (*Langmead et al., 2009*) with parameters '-q–phred33-quals -n 1 -e 99999999 l 25 -l 1 X 2000 a -m 15 S -p 6', which allows alignment of sequences with one mismatch. Expression levels of genes were quantified as transcript-per-million (TPM) values by RSEM (*Li and Dewey, 2011*) v.1.2.3 in paired-end mode. For each cell, we determined the number of genes for which at least one read was mapped, and then excluded all cells with fewer than 1,000 or more than 10,000 detected genes, or less than 25% of reads mapping to the transcriptome.

To identify variable genes a logistic regression was fit to the cellular detection fraction, using the total number of transcripts per cell as a predictor. Outliers from this curve are genes that are expressed in a lower fraction of cells than would be expected given the total number of transcripts mapping to that gene, that is, cell-type or state-specific genes. We used a threshold of deviance <−0.15, producing a set of 1910 variable genes.

### Dimensionality reduction by PCA and t-SNE

We restricted the expression matrix to the subsets of variable genes and high-quality cells noted above, and values were log$_2$-transformed, and then centered and scaled before input to PCA, which was implemented using the R function 'prcomp' from the 'stats' package. After PCA, significant principal components were identified by inspection of the scree plot. Only scores from the first 20 PCs were used as the input to further analysis.

For visualization purposes only (and not for clustering), dimensionality was further reduced using the Barnes–Hut approximate version of t-SNE (*Van Der Maaten, 2014*; *Figure 3b*). This was implemented using the 'Rtsne' function from the 'Rtsne' R package.

To identify cell types within the data, hierarchical clustering was used using the 'Ward.D2' metric in the 'hclust' R package. Genes were filtered to epithelial cell type marker genes (*Montoro et al., 2018*) before input to the clustering algorithm. Pearson's correlation was used as a distance metric. This produced three clusters, two were clearly identifiable as Basal and Club cells, based on the disjoint expression of known markers *Krt5* and *Scgb1a1*, respectively, while the third was distinguished by much lower technical quality (an average of 2373 genes detected per cell compared to 5193 for the Basal and 5480 for the club clusters, respectively, p=0.0004, Mann-Whitney U-test). These low-quality cells were not used for DE testing.

To identify the signature of moving vs non-moving basal cells (*Figure 3c*) we ran differential expression tests between cells in the Basal cluster between the two conditions (moving and non-moving), and selected genes that were differentially expressed (FDR <0.05). Differential expression tests were carried out using a two-part 'hurdle' model to control for both technical quality and mouse-to-mouse variation. This was implemented using the R package MAST (*Finak et al., 2015*), and p values for differential expression were computed using the likelihood-ratio test. Multiple hypothesis testing corrections was performed by controlling the false discovery rate using the R function 'p.adjust.'.

### Re-analysis of polidocanol injury dataset

Previously published single-cell RNA sequencing data from mouse trachea injured using polidocanol (*Plasschaert et al., 2018*) was downloaded from the NCBI GEO (GSE102580). All available unique molecular identifier (UMI) counts tables from mice at 24, 48, 72, and 168 hr after injury along with uninjured controls were downloaded. Cell types were determined using the authors provided annotations. To determine the expression of migration-associated genes in the injury response, we scored the Basal cells for the set of genes (*Figure 3c*) both significantly up- ('mover') and down-regulated ('non-mover') (*Figure 3d*). Scoring cells was computed as described previously (*Montoro et al., 2018*). To obtain a score for a specific set of n genes in a given cell, a 'background' gene set was defined to control for differences in sequencing coverage and library complexity. The background gene set was selected for similarity to the genes of interest in terms of expression level. Specifically, the 10 n nearest neighbors in the 2D space defined by mean expression and detection frequency across all cells were selected. The signature score for that cell was then defined as the mean expression of the n signature genes in that cell, minus the mean expression of the 10 n background genes in that cell.

Unsupervised cluster analysis of polidocanol-injured basal cells 24 hr after injury was computed using default settings in Seurat. Briefly, variable genes were selected using the method 'vst,' and then PCA was computed using only these genes. Shared-nearest neighbor (SNN)-based clustering was implemented using the 'FindClusters' function (resolution parameter = 0.25) using the first 25 principal components as input, resulting in two clusters (*Figure 4b*).

### Statistical analysis

Data were compared among groups using the Student's *t*-test (unpaired, two-tailed) unless otherwise specified in the Figure legends. Analysis was performed with GraphPad Prism software (version 9.1.0).

## Acknowledgements

We thank Drs. Jin-ah Park, Jeffrey Fredberg, and Jeffrey Drazen for stimulating discussion about the air-liquid-interface tissue model and migratory behaviors of cells at the beginning of this project, which was partially funded by the National Institutes of Health (NIH) via P01HL120839. We thank all of the laboratories of Dr. Jayaraj Rajagopal for providing the cell type-specific fluorescent reporter mice, access to advanced microscopy, and critical appraisal of the project. We thank NIH R01EB033155, the HSCI-CRM Flow Cytometry and Microscopy Core Facility at the Massachusetts General Hospital, and The Bauer Core Facility at Harvard University for facilitating this project.

## Additional information

#### Competing interests

Sheldon JJ Kwok: Currently an employee of and has financial interests in LASE Innovation Inc. Seok-Hyun Yun: Has financial interests in LASE Innovation Inc that were reviewed and are managed by Massachusetts General Hospital and Mass General Brigham in accordance with their conflict-of-interest policies. Vladimir Vinarsky: Currently an employee and has financial interest in Vertex Pharmaceuticals, Inc. The other authors declare that no competing interests exist.

### Funding

| Funder | Grant reference number | Author |
|---|---|---|
| National Heart, Lung, and Blood Institute | P01HL120839 | Seok-Hyun Yun |
| National Heart, Lung, and Blood Institute | F32HL154638 | Daniel T Montoro |
| National Cancer Institute | R01EB033155 | Seok-Hyun Yun |
| National Heart, Lung, and Blood Institute | K08HL124298 | Vladimir Vinarsky |

The funders had no role in study design, data collection and interpretation, or the decision to submit the work for publication.

### Author contributions

Sheldon JJ Kwok, Conceptualization, Data curation, Software, Formal analysis, Validation, Investigation, Visualization, Methodology, Writing – original draft, Project administration, Writing – review and editing; Daniel T Montoro, Conceptualization, Data curation, Software, Formal analysis, Validation, Investigation, Methodology, Writing – original draft, Writing – review and editing; Adam L Haber, Data curation, Software, Formal analysis, Investigation, Visualization, Methodology, Writing – original draft, Writing – review and editing; Seok-Hyun Yun, Conceptualization, Resources, Formal analysis, Supervision, Funding acquisition, Methodology, Writing – original draft, Project administration, Writing – review and editing; Vladimir Vinarsky, Conceptualization, Formal analysis, Funding acquisition, Investigation, Visualization, Methodology, Writing – original draft, Project administration, Writing – review and editing

### Author ORCIDs

Daniel T Montoro ⓘ http://orcid.org/0000-0002-6222-2149
Adam L Haber ⓘ http://orcid.org/0000-0002-4229-2852
Seok-Hyun Yun ⓘ http://orcid.org/0000-0002-8176-9916
Vladimir Vinarsky ⓘ http://orcid.org/0000-0003-1141-6434

### Ethics

Mice were maintained in an Association for Assessment and Accreditation of Laboratory Animal Care-accredited animal facility at the Massachusetts General Hospital, and procedures were performed with Institutional Animal Care and Use Committee (IACUC)-approved protocol 2009N000119.

### Decision letter and Author response

Decision letter https://doi.org/10.7554/eLife.76645.sa1
Author response https://doi.org/10.7554/eLife.76645.sa2

# Additional files

### Supplementary files

• Transparent reporting form

### Data availability

Sequencing data have been deposited in GEO under accession code GSE193954.

The following dataset was generated:

| Author(s) | Year | Dataset title | Dataset URL | Database and Identifier |
| --- | --- | --- | --- | --- |
| Kwok SJ, Montoro DT, Haber AL, Yun S, Vinarsky V | 2022 | Single-cell transcriptomics of dynamic cell behaviors | https://www.ncbi.nlm.nih.gov/geo/query/acc.cgi?acc=GSE193954 | NCBI Gene Expression Omnibus, GSE193954 |

The following previously published dataset was used:

| Author(s) | Year | Dataset title | Dataset URL | Database and Identifier |
| --- | --- | --- | --- | --- |
| Plasschaert LW, Zilionis R, Choo-Wing R, Savova V, Knehr J, Roma G, Klein AM, Jaffe AB | 2018 | A single cell atlas of the airway epithelium reveals the CFTR-rich pulmonary ionocyte | https://www.ncbi.nlm.nih.gov/geo/query/acc.cgi?acc=GSE102580 | NCBI Gene Expression Omnibus, GSE102580 |

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
