## [Editor Report]

This study presents a useful combination of live cell imaging with single-cell transcriptomic analyses. This is a first step to further expanding the description of cellular heterogeneity, including cellular behavior as well as gene expression profiles. The method, with more technical improvements, will be of interest to researchers who study dynamic changes in cell morphology and gene expression.

---

## [Decision Letter]

**Decision letter after peer review:**

Thank you for submitting your article "Single-cell transcriptomics of dynamic cell behaviors" for consideration by *eLife*. Your article has been reviewed by 3 peer reviewers, including Murim Choi as the Reviewing Editor and Reviewer #1, and the evaluation has been overseen by Mone Zaidi as the Senior Editor. The following individual involved in the review of your submission has agreed to reveal their identity: Martijn Nawijn (Reviewer #3).

Essential revisions:

Overall, the three reviewers agreed that the study has a novelty and will be an interest to the researchers who seek to overcome the current limitation of scRNA-seq approach. However, they also agreed that experimental limitations should be clearly defined and explained. And it should be reflected in the title and abstract. Please read the reviews carefully and address them.

*Reviewer #1 (Recommendations for the authors):*

Please discuss the potential application of the system to different tissue or disease status.

Figure 1A. On the right part, proteomics is not actually performed in the paper.

The manuscript is written in a concise way, but please briefly introduce Kaede mouse photoconversion strategy and ALI culture as they may be new to some readers.

*Reviewer #2 (Recommendations for the authors):*

1) When investigating whether the gene signature that was derived from their own data set (Figure 3c) to a mouse injury model with published data, the description of the method refers to Figure 3e, which is not presented.

2) Furthermore, the Results section refers to the injury model presented by Borthwick et al., whilst the methods section refers to the mouse injury data by Plasschaert et al.. The authors need to clarify which data set was used in Figure 4.

3) The authors should also make clear in the Results section that mini-bulks had to be constructed to retrieve the mover versus the non-mover signature and not just leave this to the methods section.

*Reviewer #3 (Recommendations for the authors):*

The single-cell transcriptomic data from this paper are not presented in any meaningful way by the authors, not even in supplementary figures. The authors claim to be able to discern clib and basal cells in non-mover regions but these data are lacking from the manuscript. The further analysis of the signatures identifying moving and non-moving (basal?) epithelial cells is also lacking, both with regard to similarity to in vivo phenotypes of the tracheal epithelial cells in the (naïve) mouse and with regard to basic functional interpretations. What sort of gene expression programs are different between these two subsets of basal cells?

The paper now reads very polished, almost like a sales brochure rather than a paper presenting a careful evaluation of the data presented. Limitations of this novel method are not mentioned at all, and the realistic implications of this new approach are also not mentioned. Instead, the authors make rather meaningless claims in the extremely short Discussion section. For instance, the first paragraph of the discussion starts with this language: "The rapid progress in spatially resolved transcriptomics is enabling the discovery and characterization of transcriptionally heterogenous cells in diverse tissue contexts (Lee et al., 2021; Ståhl et al., 2016). However, these methods do not capture the dynamics of cell behaviors that often define the unique biological processes that occur in the tissues. To address this gap, we developed an approach to examine the association of molecular and behavioral phenotypes of single cells in their native tissues." The authors can hardly claim to examine the association of behavioural phenotypes in single cells when these are analyzed at the group level only. Even more so, the 'dynamics of cell behaviour that often define the unique biological processes in tissue' (what's lacking from the existing techniques) is not really something the authors now describe. This manuscript provides a first step towards linking transcriptional and 'behavioural' phenotypes in explanted tissue – which does signify a step forward, but not the giant leap the authors seem to have in mind.

[Editors' note: further revisions were suggested prior to acceptance, as described below.]

Thank you for resubmitting your work entitled "Single-cell transcriptomics of a dynamic cell behavior in murine airways." for further consideration by *eLife*. Your revised article has been evaluated by Mone Zaidi (Senior Editor) and a Reviewing Editor.

The manuscript has been improved but there are some remaining issues that need to be addressed, as outlined below:

Specifically, one of the reviewers felt that the paper would need a thorough re-write to remove all of the unsupported claims and make a real effort to discuss the shortcomings of the model. The re-write has to be reviewed again to warrant publication in *eLife*.

*Reviewer #1 (Recommendations for the authors):*

The authors addressed the previous comments.

*Reviewer #2 (Recommendations for the authors):*

The authors have made several improvements to their manuscript which are well appreciated by this reviewer. The quality of the data presented has increased as a consequence. Unfortunately, these improvements do not extend to the discussion and interpretations offered by the authors. The manuscript still reads like a sales brochure with only positive appraisal of the latest innovative toy, rather than a balanced academic paper offering a nuanced interpretation of both the technological advances and the inherent limitations of the novel platform developed by the authors.

For instance, the authors made textual additions to the manuscript in the revision process that do not address any of the reviewers' comments and which are of limited added value except to make (bizarre) claims about the use of this platform (results (!) section, lines 89-91):

"Tissue samples from mouse disease models or human patients can also be assessed using this platform. More broadly, other types of tissues and cellular organics can also be used with this platform."

This reads very much like a claim that this platform can be used to analyse any tissue from any organism for any research question. What are cellular organics exactly?

Also the Discussion section is still severely hampered by this shortcoming: Original text also overstating the implications (discussion line 188): "a respiratory organ explant culture that maintains tissue dynamics for an extended length of time."

This system as presented here allows analysis of explanted trachea – which is not at all a respiratory organ, but merely the transport conduit for air to enter the bronchial tree and the lung. The authors show no data to support the notion that an entire mouse lung could be imaged in a meaningful way using this novel platform. Hence, such claims cannot be made.

All reviewers ask for a balanced discussion of the limitations of this platform. The authors indeed add a paragraph to the Discussion section. Unfortunately, this new paragraph only contains about 2 sentences stating the limitations, followed by another 8 sentences with unsupported claims regarding the use of their platform.

New text making unsupported claims with regard to the use of this new platform (discussion lines 203-209): "More generally, we anticipate that this live imaging-guided single-cell profiling approach will be extended to other tissues to discover underlying principles of heterogeneous cellular behaviors at homeostasis and in disease. This approach has potential application for different tissues, including cornea, esophagus, lung, prostate, kidney, intestine, and disease states such as fibrosis, cancer, and inflammation, although different organs and disease states will likely have different physiological requirements."

The authors show no data whatsoever that tissue samples from these 6 tissue types or these 3 disease states can be imaged in a meaningful way on their platform, so this type of broad claims should not be made in a scientific paper (but belong in the sales brochure for the platform). This type of language should be completely removed from the manuscript and be replaced by a discussion of the real limitations of this system: identifying cell behaviour at the single-cell level has not been achieved with this platform, and therefore cannot be linked to transcriptional phenotypes of individual cells. The data merely show single-cell transcriptomes of groups of cells with a common behaviour. In itself, this is of interest, but it needs to be presented with an accurate discussion of the limitations of the technique.

---

## [Author Response]

Reviewer #1 (Recommendations for the authors):Please discuss the potential application of the system to different tissue or disease status.Figure 1A. On the right part, proteomics is not actually performed in the paper.The manuscript is written in a concise way, but please briefly introduce Kaede mouse photoconversion strategy and ALI culture as they may be new to some readers.

Our imaging platform was designed to support ex vivo tissues and cells for long-term imaging.

However, this approach has potential application for many different tissues, including cornea, esophagus, lung, prostate, kidney, intestine, and disease states such as fibrosis, cancer, and inflammation. We have added this point as a potential application to the Discussion.

We have deleted “proteomics” from Figure 1A.

We have added an introduction and the reference for the Kaede photoconversion system in the Results section. A detailed description of the photoconversion method remains in the Methods section.

We have added a brief description of ALI culture and a reference in the Results section. More detailed description of tissue ALI culture is in the Methods section, in a subsection now named “Tracheal Explant and Tissue ALI Culture.”

Reviewer #2 (Recommendations for the authors):1) When investigating whether the gene signature that was derived from their own data set (Figure 3c) to a mouse injury model with published data, the description of the method refers to Figure 3e, which is not presented.

Thanks for identifying the typos. The reference to Figure 4b has been corrected.

2) Furthermore, the Results section refers to the injury model presented by Borthwick et al., whilst the methods section refers to the mouse injury data by Plasschaert et al.. The authors need to clarify which data set was used in Figure 4.

The reference to the appropriate dataset (Plasschaert et al., 2018) has been added.

3) The authors should also make clear in the Results section that mini-bulks had to be constructed to retrieve the mover versus the non-mover signature and not just leave this to the methods section.

To determine the mover versus non-mover signature, we did not perform any integration into ‘mini-bulk’ or ‘pseudo-bulk’ transcriptomic profiles or similar. We used information from all of the single-cell profiles for basal cells in both the moving (M) and non-moving (NM) condition and fed this into the differential expression (DE) algorithm MAST, designed specifically for single-cell RNA seq profiles, to determine genes DE between the two conditions. This approach is described in the main text: “We defined the differences in gene expression between the M basal cells and NM basal cells and identified gene signatures that are enriched (FDR < 0.05, likelihood-ratio test) either in the M or the NM basal cells (Figure 3c).” It is also added more detail in the methods section: “To identify the signature of moving vs non-moving basal cells (Figure 3c) we ran differential expression tests between cells in the Basal cluster between the two conditions (moving and non-moving), and selected genes that were differentially expressed (FDR<0.05). Differential expression tests were carried using a two part ‘hurdle’ model to control for both technical quality and mouse-to-mouse variation. This was implemented using the R package MAST (Finak et al., 2015), and P values for differential expression were computed using the likelihood-ratio test. Multiple hypothesis testing correction was performed by controlling the false discovery rate using the R function ‘p.adjust’.

Reviewer #3 (Recommendations for the authors):The single-cell transcriptomic data from this paper are not presented in any meaningful way by the authors, not even in supplementary figures. The authors claim to be able to discern clib and basal cells in non-mover regions but these data are lacking from the manuscript. The further analysis of the signatures identifying moving and non-moving (basal?) epithelial cells is also lacking, both with regard to similarity to in vivo phenotypes of the tracheal epithelial cells in the (naïve) mouse and with regard to basic functional interpretations. What sort of gene expression programs are different between these two subsets of basal cells?The paper now reads very polished, almost like a sales brochure rather than a paper presenting a careful evaluation of the data presented. Limitations of this novel method are not mentioned at all, and the realistic implications of this new approach are also not mentioned. Instead, the authors make rather meaningless claims in the extremely short Discussion section. For instance, the first paragraph of the discussion starts with this language: "The rapid progress in spatially resolved transcriptomics is enabling the discovery and characterization of transcriptionally heterogenous cells in diverse tissue contexts (Lee et al., 2021; Ståhl et al., 2016). However, these methods do not capture the dynamics of cell behaviors that often define the unique biological processes that occur in the tissues. To address this gap, we developed an approach to examine the association of molecular and behavioral phenotypes of single cells in their native tissues." The authors can hardly claim to examine the association of behavioural phenotypes in single cells when these are analyzed at the group level only. Even more so, the 'dynamics of cell behaviour that often define the unique biological processes in tissue' (what's lacking from the existing techniques) is not really something the authors now describe. This manuscript provides a first step towards linking transcriptional and 'behavioural' phenotypes in explanted tissue – which does signify a step forward, but not the giant leap the authors seem to have in mind.

We apologize that the presentation of the single-cell transcriptomics data was not made clear. Figure 3B and Figure 3C present the single-cell transcriptomic data using a dimensionality reduction (3B) and a heatmap (3C), respectively. In the revised manuscript, we clarify this further in the text and present further distinction between basal and club cells in an additional figure (new Figure 3 —figure supplement 1). The list of differentially expressed genes is included in the Figure 3 heatmap. Pathway analysis using Enrichr (Chen et al., 2013) demonstrated that the differentially expressed gene program is distinct from previously described migration programs such as EMT and unjamming.

We have added discussion on the limitations of the method in response to all 3 reviewers, including a discussion about single-cell resolution in photoconversion in this context and the use of the explant model (see end of the expanded Discussion section). We also removed the word “single” from the sentence “To address this gap, we developed an approach to examine the association of molecular and behavioral phenotypes of single cells in their native tissues” in the Discussion section.

Our aim is to bridge the divide between two powerful methodologies – cell behavioral observation through live imaging and transcriptional profiling through single cell sequencing, ultimately allowing the identification of a transcriptional signature that corresponds to that cell behavior. We agree with the reviewer that this is an important step forward towards linking dynamic cell behaviors with single cell transcriptomics and there are important challenges and limitations to resolve, and we have added this to the Discussion section.

[Editors' note: further revisions were suggested prior to acceptance, as described below.]

Reviewer #2 (Recommendations for the authors):The authors have made several improvements to their manuscript which are well appreciated by this reviewer. The quality of the data presented has increased as a consequence. Unfortunately, these improvements do not extend to the discussion and interpretations offered by the authors. The manuscript still reads like a sales brochure with only positive appraisal of the latest innovative toy, rather than a balanced academic paper offering a nuanced interpretation of both the technological advances and the inherent limitations of the novel platform developed by the authors.For instance, the authors made textual additions to the manuscript in the revision process that do not address any of the reviewers' comments and which are of limited added value except to make (bizarre) claims about the use of this platform (results (!) section, lines 89-91):"Tissue samples from mouse disease models or human patients can also be assessed using this platform. More broadly, other types of tissues and cellular organics can also be used with this platform."This reads very much like a claim that this platform can be used to analyse any tissue from any organism for any research question. What are cellular organics exactly?Also the Discussion section is still severely hampered by this shortcoming: Original text also overstating the implications (discussion line 188): "a respiratory organ explant culture that maintains tissue dynamics for an extended length of time."This system as presented here allows analysis of explanted trachea – which is not at all a respiratory organ, but merely the transport conduit for air to enter the bronchial tree and the lung. The authors show no data to support the notion that an entire mouse lung could be imaged in a meaningful way using this novel platform. Hence, such claims cannot be made.All reviewers ask for a balanced discussion of the limitations of this platform. The authors indeed add a paragraph to the Discussion section. Unfortunately, this new paragraph only contains about 2 sentences stating the limitations, followed by another 8 sentences with unsupported claims regarding the use of their platform.New text making unsupported claims with regard to the use of this new platform (discussion lines 203-209): "More generally, we anticipate that this live imaging-guided single-cell profiling approach will be extended to other tissues to discover underlying principles of heterogeneous cellular behaviors at homeostasis and in disease. This approach has potential application for different tissues, including cornea, esophagus, lung, prostate, kidney, intestine, and disease states such as fibrosis, cancer, and inflammation, although different organs and disease states will likely have different physiological requirements."The authors show no data whatsoever that tissue samples from these 6 tissue types or these 3 disease states can be imaged in a meaningful way on their platform, so this type of broad claims should not be made in a scientific paper (but belong in the sales brochure for the platform). This type of language should be completely removed from the manuscript and be replaced by a discussion of the real limitations of this system: identifying cell behaviour at the single-cell level has not been achieved with this platform, and therefore cannot be linked to transcriptional phenotypes of individual cells. The data merely show single-cell transcriptomes of groups of cells with a common behaviour. In itself, this is of interest, but it needs to be presented with an accurate discussion of the limitations of the technique.

We thank the reviewer for the detailed review of the revised manuscript and for the comments. As requested, we have added a more thorough discussion about the constraints and limitations of this method and removed the sentence referring to different tissues from the Results section. We have also clarified in the text that we studied the airway, not the entire respiratory system that would include the lung.

We note that one reviewer has commented that the manuscript “reads like a sales brochure.” This is not the intention of this manuscript and there are no patent or commercial interests that stem from this work. We hope that the latest revisions address this criticism. Several of the authors are currently working in the industry as noted in the author affiliation and in the conflicts of interest sections. As the research reported in the manuscript was performed when the authors were only affiliated with Harvard and MIT, we clarify that the industry affiliations represent the “Present address” in the author affiliations section.